# Learning gene networks underlying clinical phenotypes using SNP perturbation

**Calvin McCarter**[1], **Judie Howrylak**[2], **Seyoung Kim**[3]*

1 Machine Learning Department, Carnegie Mellon University, Pittsburgh, Pennsylvania, USA, 2 Pulmonary, Allergy and Critical Care Division, Penn State Milton S. Hershey Medical Center, Hershey, Pennsylvania, USA, 3 Computational Biology Department, Carnegie Mellon University, Pittsburgh, Pennsylvania, USA

* sssykim@cs.cmu.edu

**Data Availability Statement:** The PerturbNet software is available at https://github.com/SeyoungKimLab/PerturbNet.

**Funding:** The study was funded by NSF MCB-1149885, PA CURE, and NIH 1R21HG011116. The

## Abstract

Availability of genome sequence, molecular, and clinical phenotype data for large patient cohorts generated by recent technological advances provides an opportunity to dissect the genetic architecture of complex diseases at system level. However, previous analyses of such data have largely focused on the co-localization of SNPs associated with clinical and expression traits, each identified from genome-wide association studies and expression quantitative trait locus mapping. Thus, their description of the molecular mechanisms behind the SNPs influencing clinical phenotypes was limited to the single gene linked to the co-localized SNP. Here we introduce PerturbNet, a statistical framework for learning gene networks that modulate the influence of genetic variants on phenotypes, using genetic variants as naturally occurring perturbation of a biological system. PerturbNet uses a probabilistic graphical model to directly model the cascade of perturbation from genetic variants to the gene network to the phenotype network along with the networks at each layer of the biological system. PerturbNet learns the entire model by solving a single optimization problem with an efficient algorithm that can analyze human genome-wide data within a few hours. PerturbNet inference procedures extract a detailed description of how the gene network modulates the genetic effects on phenotypes. Using simulated and asthma data, we demonstrate that PerturbNet improves statistical power for detecting disease-linked SNPs and identifies gene networks and network modules mediating the SNP effects on traits, providing deeper insights into the underlying molecular mechanisms.

## Author summary

We describe PerturbNet, a statistical framework for learning a gene network that modulates the influence of genetic variants on phenotypes, using genetic variants as naturally occurring perturbation of a biological system. PerturbNet directly models the cascade of perturbation from genetic variants to the gene network to the phenotype network, thus integrating the existing computational tools for eQTL mapping, GWAS, co-localization analysis of eQTL and GWAS variants, and gene network discovery under SNP perturbation within a single statistical framework. We demonstrate that PerturbNet improves

funders had no role in study design, data collection and analysis, decision to publish, or preparation of the manuscript.

**Competing interests:** The authors have declared that no competing interests exist.

statistical power for detecting disease-linked SNPs and uncovers gene networks mediating the SNP effects on traits, with computational efficiency that allows for human data analysis within several hours.

## Introduction

One of the key questions in biology is how genetic variants perturb gene regulatory systems to influence disease susceptibility in population. Leveraging the naturally-occurring perturbation of gene expression by genetic variants to study gene regulatory systems has many advantages over experimental perturbation methods such as gene knockdown [1] and genome editing techniques [2], as it is more cost effective, is more easily applicable to humans, and can lead to more meaningful discoveries via subtle perturbations that occur in nature [3]. Several computational methods have been proposed to learn gene networks perturbed by single nucleotide polymorphisms (SNPs) given expression quantitative trait locus (eQTL) data [4–9]. Further combining eQTL data with clinical phenotype data to study how this gene network perturbed by SNPs in turn influences disease phenotypes has the potential to reveal the complex molecular mechanisms that explain the link between genetic variants and diseases.

However, the existing computational methods for integrating eQTL and clinical phenotype data lack the ability to learn a gene network as a mediator that receives SNP perturbation and then passes the perturbation effects onto clinical traits. Many of these methods have disregarded gene networks entirely, focusing on the co-localization of an eQTL and a trait-associated SNP [10–13], each of which was identified in a separate eQTL mapping [7, 14–16] and a genome-wide association study (GWAS) [17, 18]. Thus, their description of the regulatory role of the trait-associated SNPs was limited to the single gene whose eQTL co-localized with the GWAS SNP. Other works have used a known gene network to evaluate GWAS SNPs [19–21] or have looked for correlated associations to a trait module in a post processing step only after identifying the links between individual SNPs and traits [22, 23], thus missing out on the opportunity to discover gene networks via SNP perturbation. Bayesian networks have been used to learn gene regulatory networks and to form a predictive model for diseases with this network [24]; however, this approach relied on an elaborate pipeline of data analysis to identify disease-related gene modules and genetic variants, leading to loss of statistical power.

Here, we introduce PerturbNet, a computational framework for learning a gene network that underlies clinical traits, using genetic variants as a source of perturbation. PerturbNet unifies eQTL mapping, GWAS, co-localization analysis of eQTLs and GWAS SNPs, and gene network recovery within a single statistical framework.

PerturbNet consists of three components: a probabilistic graphical model, learning algorithms, and inference methods. The PerturbNet model represents the cascade of perturbation from SNPs to gene network and from gene network to phenotype network by stacking two sparse conditional Gaussian graphical models (sCGGMs) [4, 5], previously developed for learning gene networks perturbed by SNPs (Fig 1A). PerturbNet learning algorithm estimates the entire model from data by solving a single optimization problem that is convex with globally optimal solution, achieving high accuracy and minimal loss of statistical power. We introduce Mega-sCGGM, an sCGGM learning algorithm that is orders-of-magnitude faster with no memory restriction compared to the previous Newton coordinate descent (NCD) [25], and use it as a key module of the PerturbNet learning algorithm to allow for large human data analysis. The probabilistic graphical model framework [26] of PerturbNet naturally leads to a set of inference methods for revealing perturbations and regulatory relationships that are not

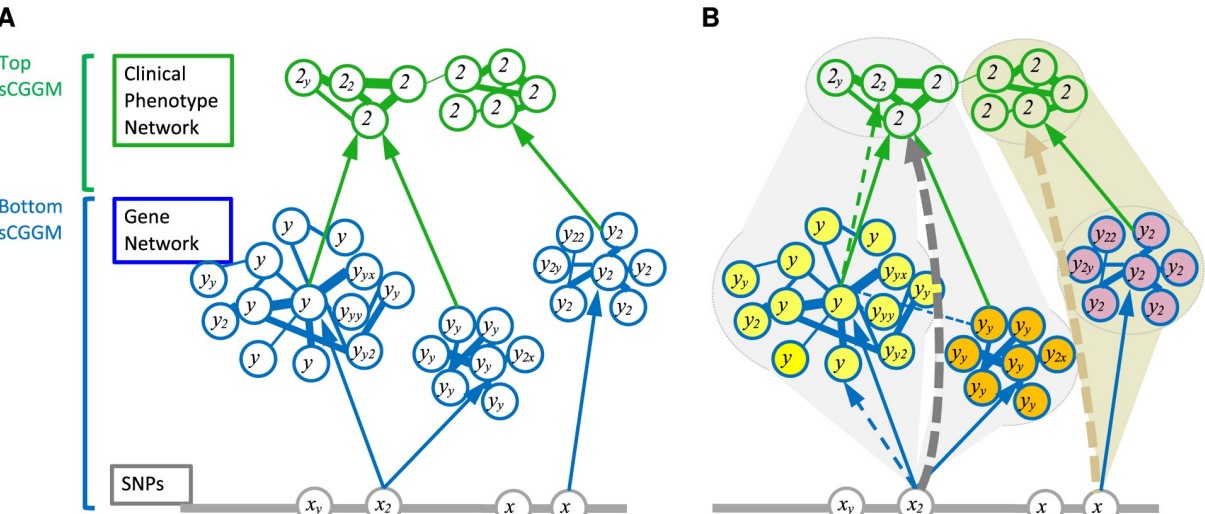

**Fig 1. Overview of PerturbNet.** (A) PerturbNet model as a cascade of two sCGGMs. The blue sCGGM for $p(\mathbf{y}|\mathbf{x})$ models a gene network perturbed by SNPs and the green sCGGM for $p(\mathbf{z}|\mathbf{y})$ models a clinical trait network perturbed by gene expression levels. (B) PerturbNet inference methods. Given the PerturbNet model in (A), the PerturbNet inference methods infer SNP effects on phenotypes mediated by the gene network (gray and tan dashed arrows), a decomposition of SNP effects on phenotypes into component SNP effects mediated by each gene module (yellow and orange modules for gray dashed arrow and pink module for tan dashed arrow), and the posterior gene network with additional edges between the gene modules influencing the same phenotypes (blue dashed edge). PerturbNet also inherits the inference methods of sCGGMs for inferring indirect perturbation effects resulting from the direct perturbation effects propagating through the networks (blue and green dashed arrows).

explicitly represented as edges but only implicitly present in the model (Fig 1B). Given the model estimated from data, PerturbNet uses these inference methods to characterize how the gene network and modules mediate the SNP effects on clinical traits, uncovering the molecular mechanisms that underlie the link between SNPs and traits.

We demonstrate PerturbNet using simulated and patient cohort data from the Childhood Asthma Management Program (CAMP) [27–29]. Compared to the existing methods that combine eQTL and GWAS data [12, 13], we show PerturbNet provides higher sensitivities for detecting trait-perturbing variants and for identifying genes that mediate the genetic effects on traits, as it learns and leverages gene networks rather than individual genes as mediators. We show the PerturbNet inference methods extract rich information on the role of the gene networks as mediators.

## Results

### Methods overview

We briefly describe the model, learning algorithms, and inference methods for PerturbNet (see Methods for detail). The PerturbNet model represents the cascaded perturbation from SNPs to gene network to trait network as a Gaussian chain graph model obtained by threading two sCGGMs, each modeling a network and perturbation of this network at the given layer (Fig 1A). The PerturbNet learning algorithm obtains a sparse estimate of the model with few edges in the network by minimizing the negative log-likelihood of data with $L_1$-regularization. Since the PerturbNet learning algorithm uses an sCGGM learning method as a key module, we introduce Mega-sCGGM, an efficient sCGGM learning algorithms for reducing computation time and for removing memory constraint of the previous algorithm.

PerturbNet provides three inference methods for extracting detailed information on the cascaded network perturbation from the estimated model: INF-I for inferring SNP effects on

traits mediated by gene network (gray and tan dashed arrows in Fig 1B), which in turn can be used to reveal the co-localization of the inferred trait-associated SNPs and the eQTLs; INF-II for decomposing SNP effects on traits into components mediated by each gene module (yellow, orange, and pink modules in Fig 1B) within the network, uncovering the mediator genes and modules for SNP effects on traits and revealing the pathways underlying these SNP-trait links; and INF-III for obtaining the posterior gene network after accounting for correlated clinical traits (blue dashed edge in Fig 1B). Additionally, PerturbNet inherits the inference methods of sCGGMs [5] for inferring indirect perturbation effects that arise from another direct perturbation exerting its influence through the network (blue and green dashed edges in Fig 1B).

Using simulated data and asthmatic patient cohort data from the CAMP [29], we evaluate PerturbNet against eCAVIAR [13], PrediXcan [12], and two-layer Lasso (S1 Text). All of these previous methods combined eQTL and GWAS data to identify genes underlying GWAS SNPs, but none considered a gene network as a mediator of SNP effects on traits.

## Simulation experiments

Using data simulated from real genotypes in the CAMP study and the known ground-truth models, we evaluated PerturbNet and other methods on the accuracy of detecting trait-perturbing SNPs and mediator genes, and on the accuracy of gene network recovery (see Methods). We simulated datasets with 5, 000 genes and 100 clinical traits, given 10, 000 SNPs on chromosome 1 of 540 non-Hispanic Caucasians from the CAMP study.

For the tasks of identifying trait-perturbing SNPs and mediator genes, PerturbNet had the highest sensitivities across all false discovery rates (FDRs), regardless of the model types used to simulate data, and regardless of whether a gene is a mediator acting independently or cooperating with other genes in the network Figs (2A–2C). Unlike other methods, PerturbNet has the ability to distinguish between direct and indirect perturbation of a network. This enables a categorization of the role of the mediator gene into four possible combinations, based on whether the perturbation effect the mediator gene receives from SNPs and passes onto clinical trait network is direct or indirect. PerturbNet identified these categories for each mediating gene with high accuracy (Fig 2B).

On gene network recovery, PerturbNet had higher accuracy on data simulated with SNP perturbation, compared to sparse Gaussian graphical models (GGMs) [30, 31], a popular statistical method for learning gene networks (Fig 2D). PerturbNet's accuracy did not suffer on data simulated without SNP perturbation. PerturbNet achieved higher accuracy by estimating the entire model of cascaded network perturbation in a single statistical analysis.

## Analysis of CAMP data

**PerturbNet is scalable for human data analysis.**   We compared the computation time of PerturbNet against eCAVIAR, PrediXcan, and two-layer Lasso. We also compared the scalability of Mega-sCGGM, the key subroutine of PerturbNet, against Lasso and NCD [25], the previous state-of-the-art optimization method for learning an sCGGM. We used all CAMP data for 11,598 genes and 35 phenotypes, while varying the number of SNPs from 40,056 SNPs in chromosome 1 to 212,757 SNPs in chromosomes 1-6 and to 495,597 SNPs in all autosomal chromosomes.

PerturbNet analyzed all CAMP data in less than four hours and scaled similarly to other methods that are not concerned with gene networks (Fig 3A). On expression and SNP data, Mega-sCGGM scaled similarly to Lasso [32], a computationally efficient but less powerful method, as it only learns eQTLs but not the gene network (Fig 3B). Mega-sCGGM was also

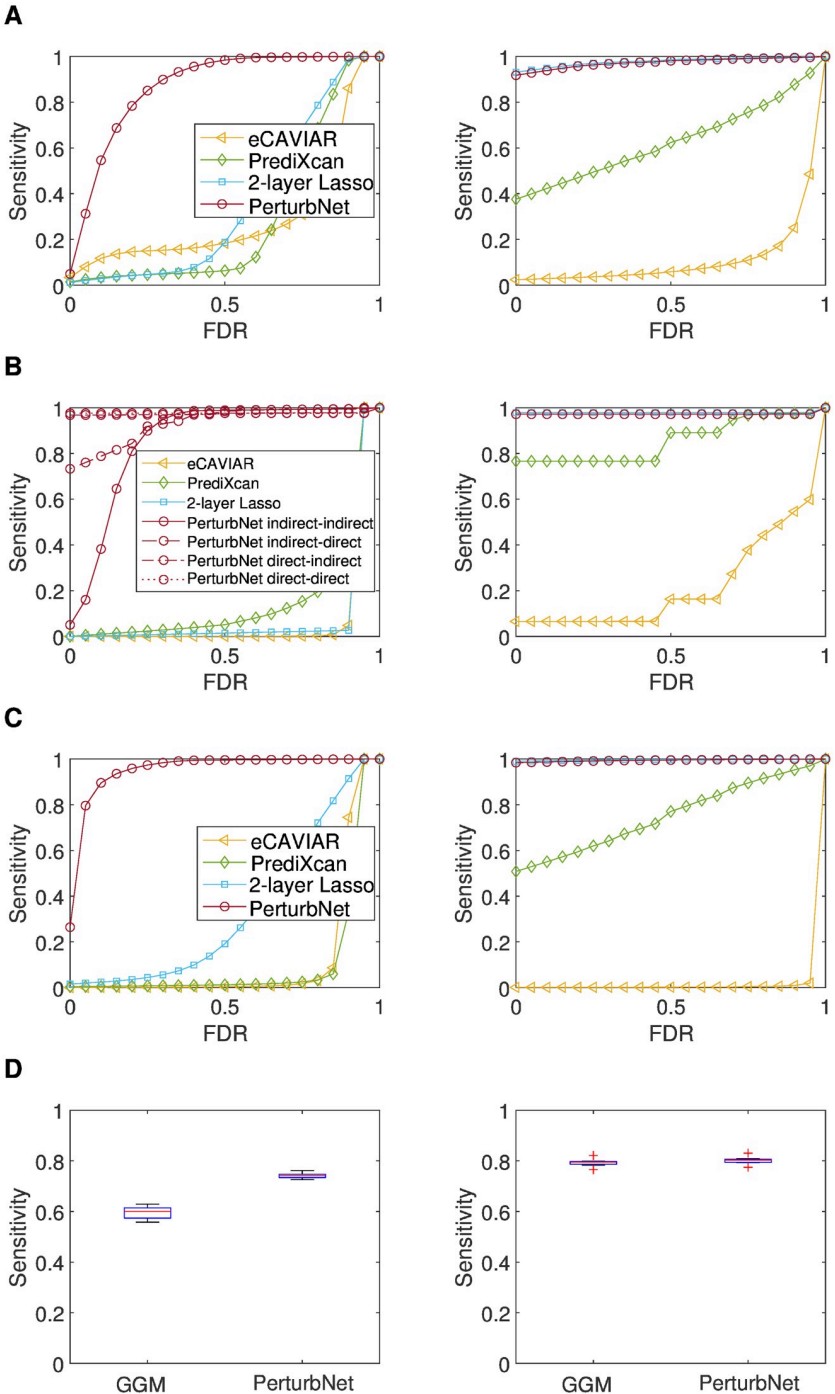

**Fig 2. Results on simulated data.** The accuracy of PerturbNet, eCAVIAR, PrediXcan, and two-layer Lasso is shown for (A) the recovery of SNPs perturbing traits, (B) the recovery of genes mediating the effect of each SNP on each trait (for PerturbNet, categories for the role of each mediator gene are shown for four possible combinations of direct vs. indirect perturbations that the mediator gene receives from SNPs and passes onto clinical trait network), and (C) the recovery of genes mediating the overall SNP effects on each trait. Ground-truth models with two-layer sCGGMs (left) and two-layer linear regression models (right) were used to simulate data. (D) The accuracy of PerturbNet and GGM on gene network recovery. Networks with SNP perturbation (left) and without SNP perturbation (right) were used as the ground-truth models. Sensitivities at FDR = 0.05 are shown.

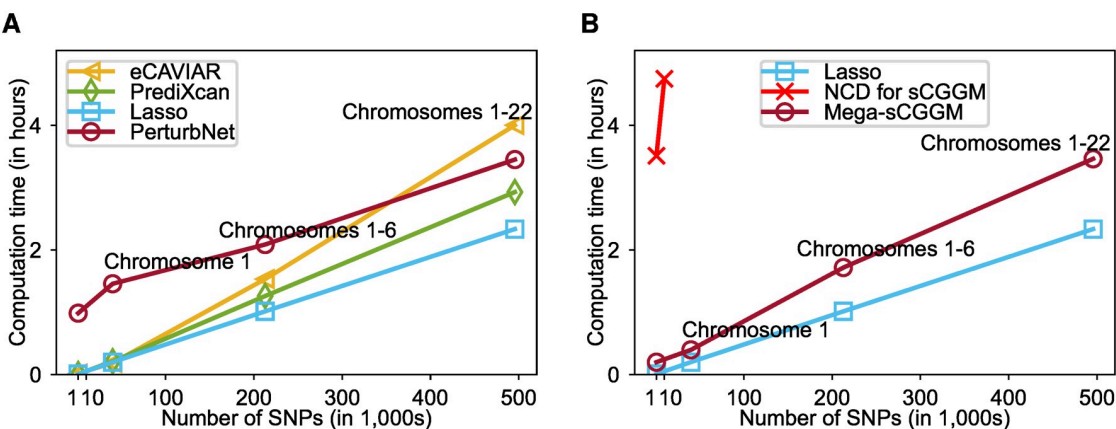

**Fig 3. Comparison of computation time.** (A) Computation time for analysis of SNP, expression, and clinical trait data from the CAMP study. (B) Computation time for analysis of eQTL data from the CAMP study. NCD, the previous algorithm for sCGGMs, ran out of memory at chromosome 1. The expression levels of 11,598 genes, 35 traits, and varying numbers of SNPs were used.

significantly more efficient than NCD in terms of both memory requirement and computation time: it analyzed all 495, 597 SNPs within three and a half hours, whereas NCD spent the same amount of time on 1,000 SNPs and ran out of memory on SNPs from chromosome 1. When an estimate from NCD could be obtained on the dataset with 1, 000 SNPs, NCD and Mega-sCGGM obtained an identical estimate, as was expected since they solve the same convex optimization problem with a single global optimum using an exact method without approximation.

**PerturbNet asthma gene network and gene modules.** PerturbNet has the unique ability to determine the gene network perturbed by SNPs underlying clinical traits. To demonstrate this, we first examine the PerturbNet model estimated from the CAMP data (S1 Fig), focusing on the asthma gene network. In the gene network, out of 11,598 genes, 6,102 genes had at least one neighbor in the network, while the rest of the genes were singleton nodes in the network. To characterize the module structure in this network for further functional analysis, we post-processed the estimated gene network by excluding the singleton nodes and clustering the part of the network corresponding to the 6,102 genes into 20 modules, using the $k$-way network clustering algorithm METIS [33]. We tried a variety of settings for the number of clusters $k$ and chose $k = 20$ modules as it produced modules that map to dense subnetworks within the full network from visual inspection (S1(B) Table).

A close examination of the 20 modules in the PerturbNet gene network revealed that a sub-set of those modules, modules 13-20, are likely to be involved in asthma. To determine the functional role of the 20 gene modules, we performed GO gene set enrichment analysis [34]. For each module, we performed a Fisher's exact test to find significantly enriched GO categories in biological processes ($p$-value $< 0.05$ after Bonferroni correction for multiple testing), using the GO database with annotations for 21,002 genes. Modules 13-15 had statistically significant enrichment of GO terms related to immune system function (S1 Table), suggesting these modules are likely to play a role in asthma, since asthma is an immune disorder. Even though modules 16-20 did not have any significant enrichment of the same GO categories, the set of 374 genes in these modules that are connected to genes in modules 13-15 in the posterior gene network from the PerturbNet inference method INF-III (S2 Fig) were significantly enriched for several GO categories related to immune system processes ($p$-value $< 0.05$ after Bonferroni correction; S2 Table). These genes with immune system GO annotations formed sub-clusters within each of modules 13-20. Overall, the enrichment of immune-related genes

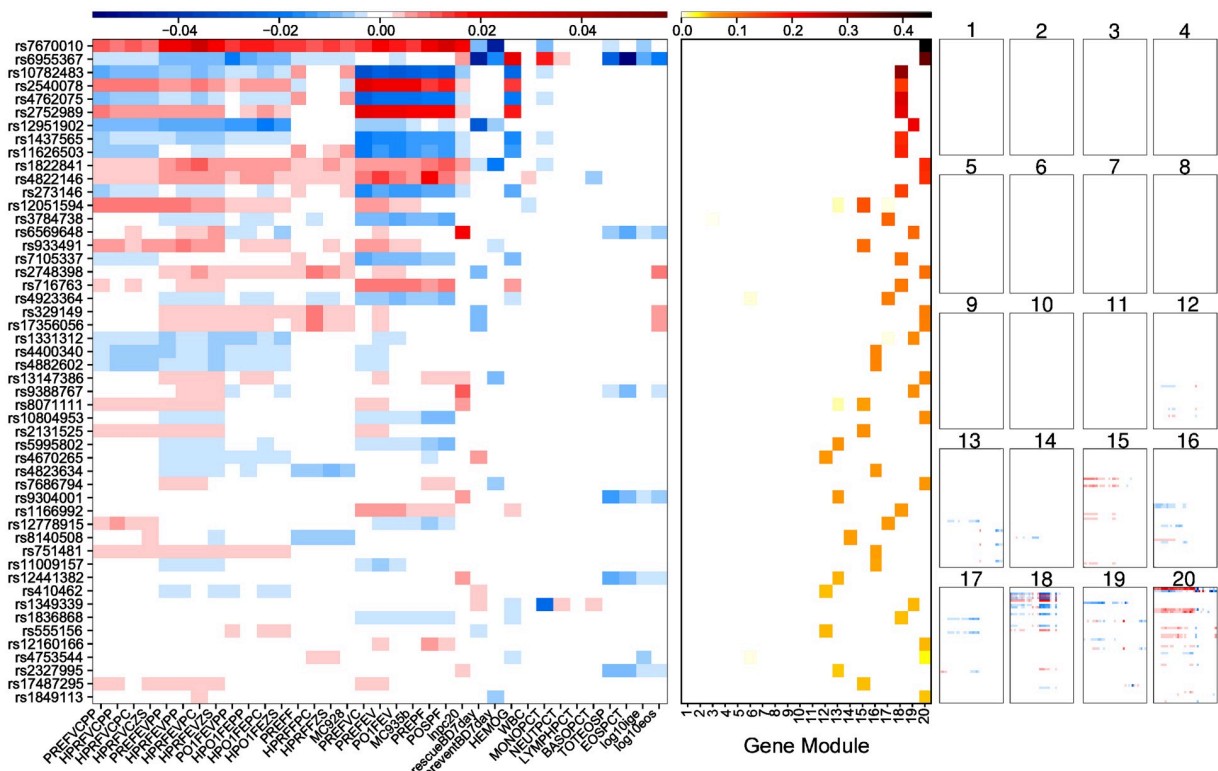

**Fig 4. SNP effects on asthma traits mediated by gene network and gene modules from PerturbNet.** SNP effects on traits for top 50 SNPs from the PerturbNet inference method INF-I (left). The decomposition of these SNP effects into 20 component effects mediated by each gene module from the PerturbNet inference method INF-II (right). Summary of these component effects by summing over traits (center).

in modules 13-20 suggests that these modules are likely the network components involved in the disease process of asthma.

Furthermore, in the overall PerturbNet model, while all gene modules were perturbed by SNPs in the SNP-to-gene sCGGM (S3(A) Fig), modules 13-20 had substantially larger effects on the asthma phenotypes in the gene-to-trait sCGGM (S3(B) Fig), providing additional evidence that modules 13-20 are likely to be implicated in asthma.

**PerturbNet finds gene modules 13-20 mediate most of the genetic effects on asthma traits.** Another unique feature of PerturbNet is its inference methods for uncovering how SNP effects on traits are mediated by the gene network and modules. Given the estimated PerturbNet model, we first applied the inference method INF-I to infer the genetic effects on traits mediated by the network, identifying top 500 SNPs with the largest overall effects on traits (S4 Fig and Fig 4 left).

Then we applied INF-II to infer the decomposition of these SNP effects from INF-I into component effects mediated by each of the 20 gene modules (Fig 4 right). For 99 out of top 500 SNPs, the primary mediators were modules rather than single isolated genes in the network. Moreover, for all of those 99 SNPs, modules 13-20 mediated nearly all of those genetic effects (Fig 4 right), providing evidence that the modules enriched with genes involved in immune system function above (S1 and S2 Tables, S3 Fig) also explain the molecular mechanisms behind the top asthma SNPs. Additionally, the effect of each SNP on traits was mediated by only one of modules 13-20 (Fig 4 center), except for two SNPs with two mediator modules, suggesting a SNP influences asthma traits via localized perturbation of the gene network. Our

results demonstrate that PerturbNet inference methods can identify candidate gene modules that are mediators of SNP effects on traits. Unlike eCAVIAR, PerturbNet explains the molecular mechanisms behind trait-associated SNPs not simply through an individual gene, whose eQTLs co-localize with GWAS SNPs, but through gene modules mediating SNP effects on traits.

Most of these asthma SNPs from the inference co-localized with eQTLs for modules 13-20 in the estimated SNP-to-gene sCGGM component of the PerturbNet model (S5 Fig). Although a small subset of these trait-perturbing SNPs co-localized with eQTLs for modules 1-12, the genetic effects mediated by these modules were weak, suggesting these modules do not propagate the SNP perturbations to the traits and are not likely to play a central role in the development of the disease.

**PerturbNet SNPs are most strongly supported by external annotations.** For the top asthma-associated SNPs identified by PerturbNet, eCAVIAR, PrediXcan, and two-layer Lasso, we examined the functional annotations in three external resources (see Methods): the Encyclopedia of DNA Elements (ENCODE) DNase hypersensitivity sites (GSM1008572) in GM12878 B-lymphoblastoid cell line [35], a blood cell type as in CD4+ T lymphocytes in the CAMP study; RegulomeDB [36] integrating ENCODE and other sources of annotations for diverse cell types; and SNPs in the National Human Genome Research Institute (NHGRI) GWAS catalog.

PerturbNet SNPs were most strongly supported as functional by all of the three external resources. PerturbNet SNPs had the largest overlap with the DNase-seq hypersensitivity sites from the ENCODE B-lymphoblastoid cell-line [35] with statistical significance (Fig 5A), and also with functionally annotated SNPs in RegulomeDB [36], especially with SNPs in the RegulomeDB category 1, which corresponds to the strongest evidence of being located in the functional region and includes SNPs previously identified as eQTLs (Fig 5B). Furthermore, PerturbNet had the largest numbers of SNPs located within 10, 20, 50, and 100kb of 557 asthma SNPs in the NHGRI GWAS catalog [37] (Fig 5C and S7 Fig). Applying the PerturbNet inference method to the PerturbNet asthma SNPs within 20kb of the NHGRI GWAS SNPs revealed for all except for one SNP the SNP effects on traits were mediated by modules 13-20 (Fig 5D), showing they may be the key modules underlying these SNPs from the GWAS catalog.

**Module 13 mediates the genetic effects of SNP rs12441382 on asthma traits.** We examined module 13 mediating the perturbation of traits by SNP rs12441382, one of the top PerturbNet SNPs (Fig 6). This SNP is located 17kb from rs1841128 in the NHGRI catalog [37] for a lung function trait, forced vital capacity [38]. These two SNPs have normalized linkage disequilibrium coefficient $D' = 1.0$ in the 1000 Genomes Project CEU population [39]. PerturbNet found two genes, *ATF3* and *EGR2*, that are directly perturbed by this SNP with the strongest effect sizes. Both genes have been previously linked to allergic asthma. *ATF3* is a known negative regulator of allergic asthma and was recently proposed to be a hub of the cellular adaptive-response network, playing a key role in immune diseases [40]. The perturbation of *ATF3* by this SNP is also reflected in statistically significant association in univariate regression analysis (FDR $q = 6.851 \times 10^{-5}$). *EGR2* has been linked to migration of CD4+ T cells to lung and to blood eosinophil levels in asthma [41–43]. Indirectly perturbed genes near *ATF3* and *EGR2* in the network include *C2* [44–46], *SERPING1* [47], *ZG16B* [48], and *CEACAM3* [49], all of which have been linked to immune response, asthma, or auto-immune diseases. PerturbNet provided new insights into the molecular characterization based on *ATF3*-centered pathway for the known GWAS SNP whose functional role in asthma has not been fully elucidated.

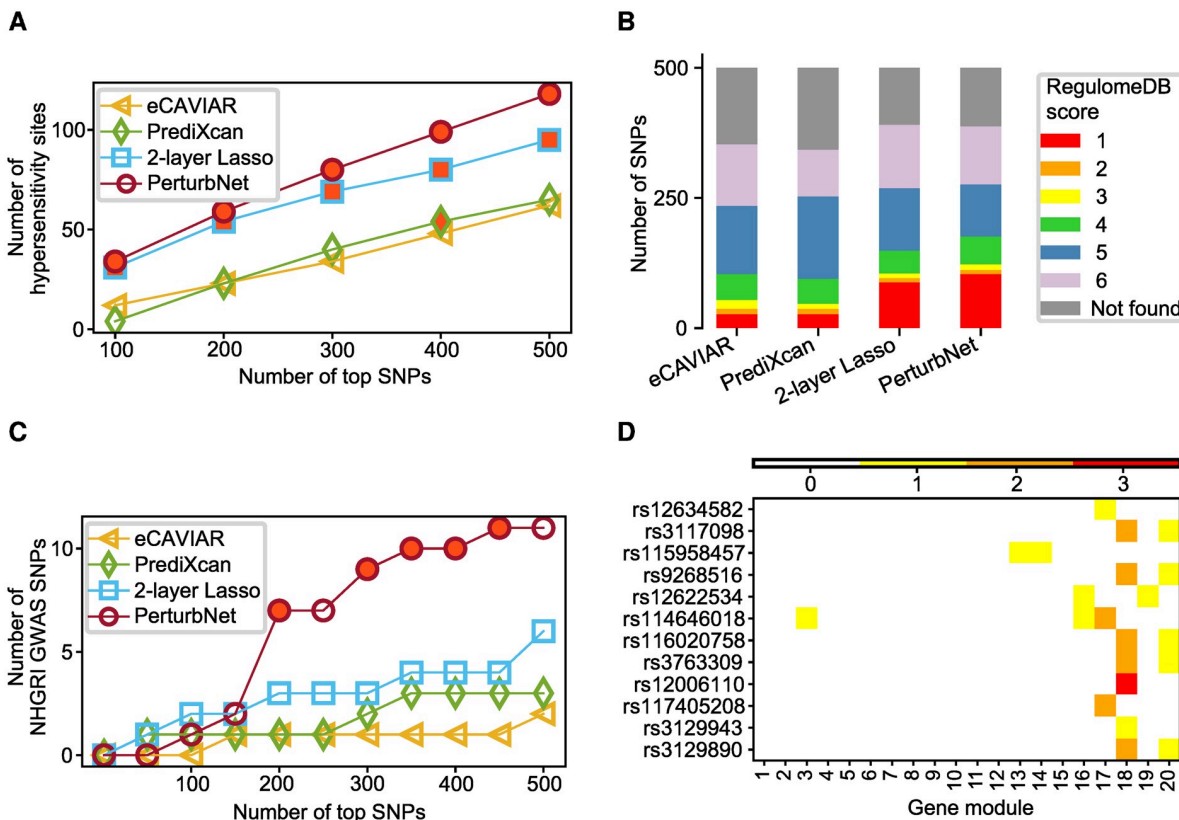

**Fig 5. Comparison of top asthma SNPs from different methods.** Top asthma SNPs identified by PerturbNet, eCAVIAR, PrediXcan, and two-layer Lasso are compared against external annotations. (A) Overlaps between top asthma SNPs from each method and the ENCODE DNase hypersensitivity sites for B-lymphoblastoid cell line. Statistically significant enrichments ($p$-value $< 0.05$) are highlighted with red. (B) Overlaps between top 500 asthma SNPs from each method and functionally annotated SNPs in RegulomeDB. Lower scores indicate more likely to be functional. (C) Overlaps with asthma SNPs in the NHGRI GWAS catalog. An overlap is defined as less than 20kb between two SNPs. Statistically significant enrichments (permutation test $p$-value $< 0.05$) are highlighted with red. (D) PerturbNet gene modules mediating the effects of NHGRI GWAS SNPs that overlap with PerturbNet asthma SNPs in (C). The colorbar indicates the number of PerturbNet asthma SNPs overlapping with each NHGRI SNP.

**PerturbNet robustness analysis on asthma data.** We evaluated the robustness of PerturbNet by fitting the model to perturbed CAMP datasets with either fewer samples or samples with added noise. To generate a modified CAMP dataset with fewer samples, given a desired fraction of samples, we took a random subset of samples without replacement of the fully-observed samples and of the partially-observed samples. To generate CAMP datasets with added noise, given a desired noise standard deviation, we added independent Gaussian noise to both gene expression and phenotype data. For each of the modified datasets, we fit a PerturbNet model. Then, we evaluated the nonzero elements of the estimated parameters in terms of the area under curve (AUC) of receiver operating characteristic (ROC) curves, assuming the parameters obtained from the original CAMP dataset is the ground-truth. We found that the PerturbNet estimates tend to be robust across different fractions of samples and noise levels and do not show significant changes with small perturbation in data (Fig 7).

## Discussion

PerturbNet combined genetic variant, expression, and trait data in a single statistical analysis to model the cascade of influence from SNPs to gene network to phenotype network in a

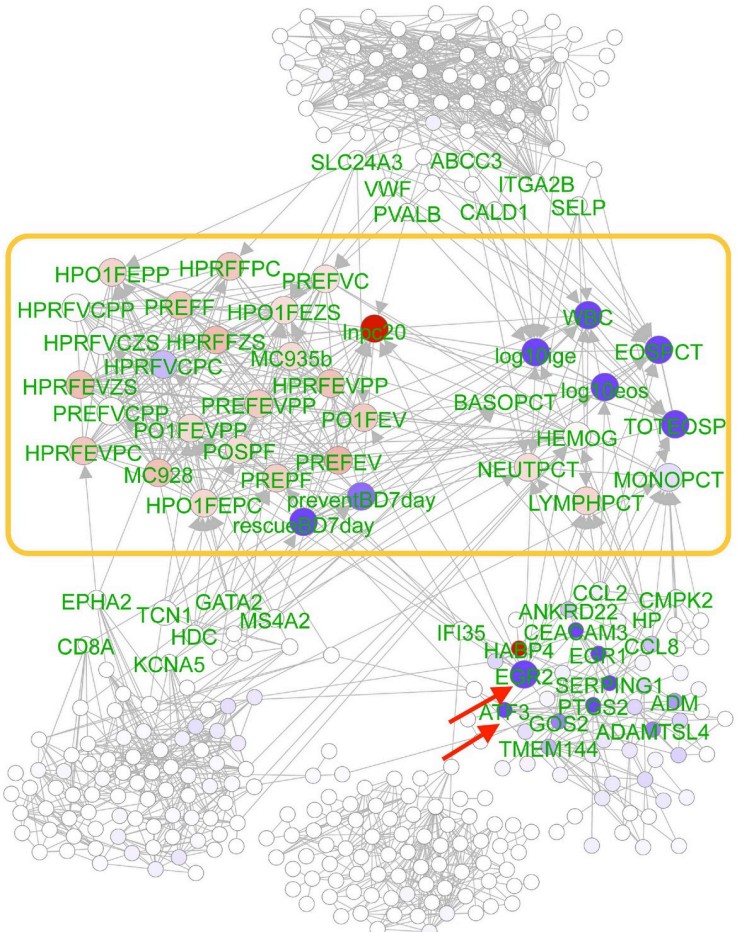

**Fig 6. PerturbNet module 13 mediating the effects of SNP rs12441382 on asthma traits.** The PerturbNet cascade from SNP rs12441382 to module 13 (outside of yellow box) to trait network (inside of yellow box). Direct SNP perturbations (red arrows), indirect SNP perturbations of expression levels and phenotypes (node colors), and the component of the SNP effects mediated by each gene, summed across all traits (gene node sizes) are shown.

multi-layered biological system. Our results demonstrated several key advantages of our approach. First, PerturbNet detected SNPs perturbing traits through gene expression with higher sensitivity by leveraging gene networks, compared to the existing methods that narrowly focused on detecting the co-localization of eQTLs and GWAS variants with multi-stage analysis [10–13]. Second, unlike other methods, PerturbNet, using inference methods, could infer how different parts of the gene network act as mediators of the genetic influence on phenotypes, providing insights into the gene regulatory mechanisms underlying the genetic effects on the phenotypes. Finally, PerturbNet accomplished these without sacrificing the computation time, allowing for human data analysis within a few hours.

PerturbNet is a flexible tool that can be easily extended to model a more complex biological system under genetic influence. Because PerturbNet uses sCGGMs as building blocks, sCGGMs could be threaded in different ways to integrate different data types. To model tissue-specific gene networks underlying genetic effects on clinical traits, a PerturbNet model could be set up with multiple SNP-to-gene sCGGMs, one for each tissue type, all of which are linked to the same gene-to-trait sCGGM. Such a model could explain the phenotypic variability by the gene expression of the relevant tissue types and enable a discovery of SNPs that affect

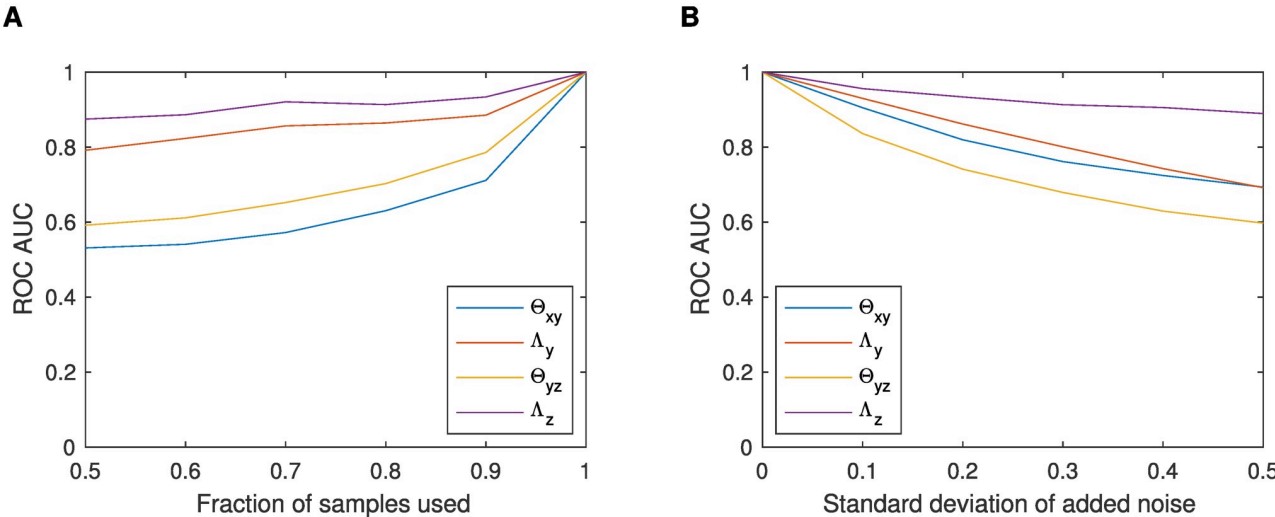

**Fig 7. Robustness of the PerturbNet estimates on perturbed CAMP datasets.** ROC AUCs are shown for PerturbNet models fit to the CAMP datasets modified with (A) fractions of samples and (B) noise added to the expression and clinical trait data. The non-zero elements in the PerturbNet parameters obtained from the modified datasets were compared against those from the original CAMP data.

disease phenotypes through specific tissue types. To model genetic perturbation of a biological system with more than two layers that includes microRNAs, proteomes, and metabolomes, multiple sCGGMs could be combined in a PerturbNet model by assigning an sCGGM to each level of the cascaded perturbation. While in this work, we only considered continuous-valued traits, discrete-valued traits could be modeled with conditional random fields (CRFs) [26, 50], a discrete counterpart of sCGGMs, where efficient learning and inference algorithms are available from the huge literature on applications of CRFs to image and text data analysis.

Our study considered genetic variant, expression, and clinical data collected from the same patient cohort. One future direction is to apply PerturbNet to datasets, where different data types were collected from different cohorts. Each component sCGGM in the model could be estimated using data collected from different cohorts, if not all data types are available from the same cohort.

Another future direction is to extend PerturbNet to account for the characteristics of non-independent samples such as population structure. Linear mixed effects models (LMMs) have been widely used in genetics to model non-homogenous samples with population stratification. Multi-trait mixed effects models have been proposed to extend an LMM to model multivariate traits through matrix normal distribution [51]. A similar strategy could be used to combine LMMs with sCGGMs in a PerturbNet model.

Finally, PerturbNet could be extended to model non-linear relationships among expression/clinical traits and between traits genetic variants. The sCGGM component of PerturbNet assumed that multivariate expression traits are Gaussian distributed. This is essentially equivalent to assuming that the expression of each gene is a linear function of genetic variants and the expression of other neighboring genes in the gene network. Gaussian graphical models have been extended to non-parametric models to model non-linear dependencies among multiple random variables in high-dimensional setting [52]. Our future work includes employing a similar strategy to extend sCGGMs of PerturbNet to model non-linear dependencies.

The PerturbNet software is available at https://github.com/SeyoungKimLab/PerturbNet.

## Methods

We describe the probabilistic graphical model, learning algorithms, and inference methods of our PerturbNet framework.

### PerturbNet model

We introduce the probabilistic graphical model of the PerturbNet framework. Let $\mathbf{x}$ denote genotypes at $p$ loci for an individual, given as the number of minor alleles at the loci taking values from {0, 1, 2}. Let $\mathbf{y} \in \mathbb{R}^q$ denote expression levels for $q$ genes and $\mathbf{z} \in \mathbb{R}^r$ measurements for $r$ phenotypes for the same individual. Then, PerturbNet models the cascade of influence from SNPs to a gene network to a phenotype network as a Gaussian chain graph model (Fig 1A), a factorized conditional probability distribution defined as follows:

$$p(\mathbf{y}, \mathbf{z}|\mathbf{x}) = p(\mathbf{y}|\mathbf{x})p(\mathbf{z}|\mathbf{y}). \tag{1}$$

Each probability factor above is modeled as an sCGGM [4, 5, 25]:

$$p(\mathbf{y}|\mathbf{x}) = \exp\left(-\frac{1}{2}\mathbf{y}^T\mathbf{\Lambda_y}\mathbf{y} - \mathbf{x}^T\mathbf{\Theta_{xy}}\mathbf{y}\right)/Z_1(\mathbf{x}), \tag{2}$$

$$p(\mathbf{z}|\mathbf{y}) = \exp\left(-\frac{1}{2}\mathbf{z}^T\mathbf{\Lambda_z}\mathbf{z} - \mathbf{y}^T\mathbf{\Theta_{yz}}\mathbf{z}\right)/Z_2(\mathbf{y}). \tag{3}$$

The first probability factor in Eq (2) models the gene network perturbed by SNPs, representing the gene network as a $q \times q$ positive definite matrix $\mathbf{\Lambda_y}$ and the SNP perturbation of this network as $\mathbf{\Theta_{xy}} \in \mathbb{R}^{p \times q}$. The second probability factor in Eq (3) models the phenotype network perturbed by gene expression levels, where the phenotype network $\mathbf{\Lambda_z}$ is an $r \times r$ positive definite matrix and the perturbation of this network by gene expression levels is modeled as $\mathbf{\Theta_{yz}} \in \mathbb{R}^{q \times r}$. $Z_1(\mathbf{x})$ and $Z_2(\mathbf{y})$ in Eqs (2) and (3) are constants for ensuring that each sCGGM is a proper probability distribution that integrates to one. Our model in Eq (1) defines a probability distribution over the graph shown in Fig 1A. A non-zero value in the $(i, j)$th element of the network parameters, $[\mathbf{\Lambda_y}]_{i,j}$ of $\mathbf{\Lambda_y}$ and $[\mathbf{\Lambda_z}]_{i,j}$ of $\mathbf{\Lambda_z}$, corresponds to presence of an edge between the $i$th and $j$th nodes of the corresponding network. Similarly, a non-zero value in the $(i, j)$th element of the perturbation parameters, $[\mathbf{\Theta_{xy}}]_{i,j}$ of $[\mathbf{\Theta_{xy}}]$ and $[\mathbf{\Theta_{yz}}]_{i,j}$ of $[\mathbf{\Theta_{yz}}]$, indicates an edge between the $i$th perturbant and the $j$th node in the network.

This Gaussian chain graph model corresponds to the continuous counterpart of the chain graph model obtained by threading CRFs for discrete random variables [50]. CRFs and the chain graph models built from CRFs have been hugely popular in other application areas of statistical machine learning such as text modeling and image analysis for modeling multiple correlated output features influenced by input features [26, 50, 53]. Here, we explore the use of a chain graph model constructed with sCGGMs. We develop an efficient learning algorithm that runs on human data within a few hours and a set of inference algorithms for dissecting the gene regulatory mechanisms that govern the influence of SNPs on phenotypes.

### PerturbNet inference methods

The probabilistic graphical model for PerturbNet naturally leads to a set of inference methods for revealing dependencies that are not explicitly represented as edges in the model. In general, inference methods in probabilistic graphical models are computationally expensive [26].

However, because the PerturbNet model is built upon sCGGMs, a form of Gaussian distributions, it is possible to obtain efficient inference methods that involve simple matrix operations. Below, we describe two inference methods that PerturbNet inherits from sCGGM and introduce three inference methods for the Gaussian chain graph model of PerturbNet.

PerturbNet inherits the following two inference methods directly from the inference method for an sCGGM [4, 5]. They are used to infer the indirect perturbation effects that arise from the direct perturbation effects propagating to other parts of the network.

- **Indirect SNP perturbation effects on gene expression levels:** $\mathbf{B_{xy}} = -\mathbf{\Theta_{xy}}\mathbf{\Lambda_y}^{-1}$, where $[\mathbf{B_{xy}}]_{k,i}$ represents the indirect perturbation effect of SNP $k$ on the expression level of gene $i$ (Fig 1B, blue dashed arrow). This can be seen by re-writing the sCGGM component $p(\mathbf{y}|\mathbf{x})$ of the PerturbNet model as follows:

$$p(\mathbf{y}|\mathbf{x}) = N(\mathbf{B_{xy}}^T\mathbf{x}, \mathbf{\Lambda_y}^{-1}),\tag{4}$$

from which the marginal distribution for the expression level $[\mathbf{y}]_i$ of gene $i$ can be obtained as $p([\mathbf{y}]_i|\mathbf{x}) = N([[\mathbf{B_{xy}}]_{:,i}]^T\mathbf{x}, [\mathbf{\Lambda_y}^{-1}]_{i,i})$, where $[\mathbf{B_{xy}}]_{:,i}$ represents the $i$th column of $\mathbf{B_{xy}}$. While $[\mathbf{\Theta_{xy}}]_{k,i}$ models the direct perturbation effect of SNP $k$ on the expression of gene $i$, $[\mathbf{B_{xy}}]_{k,i}$ corresponds to the overall perturbation effect that aggregates all indirect influence of SNP $k$ on gene $i$ through other genes. When SNP $k$ does not influence the expression of gene $i$ directly but exerts influence on gene $i$ through other genes connected to gene $i$ in the network $\mathbf{\Lambda_y}$, we have $[\mathbf{\Theta_{xy}}]_{k,i} = 0$ but $[\mathbf{B_{xy}}]_{k,i} \neq 0$.

- **Indirect effects of gene expression levels on clinical phenotypes:** $\mathbf{B_{yz}} = -\mathbf{\Theta_{yz}}\mathbf{\Lambda_z}^{-1}$, where $[\mathbf{B_{yz}}]_{k,i}$ represents the indirect influence of the expression level of gene $k$ on phenotype $i$ (Fig 1B, green dashed arrow). Similarly as above, this can be seen by deriving the marginal distribution from the sCGGM component $p(\mathbf{z}|\mathbf{y})$ of the PerturbNet model as follows:

$$p(\mathbf{z}|\mathbf{y}) = N(\mathbf{B_{yz}}^T\mathbf{y}, \mathbf{\Lambda_z}^{-1}).$$

Then, the marginal distribution for $[\mathbf{z}]_i$ of phenotype $i$ can be obtained as $p([\mathbf{z}]_i|\mathbf{y}) = N([[\mathbf{B_{yz}}]_{:,i}]^T\mathbf{y}, [\mathbf{\Lambda_z}^{-1}]_{i,i})$.

PerturbNet builds upon the sCGGM inference methods above and provides the following three inference methods on the sparse Gaussian chain graph model for characterizing the role of the gene network as a mediator of SNP effects on clinical traits.

- **INF-I for inferring SNP effects on clinical phenotypes mediated by gene network:** INF-I computes $\mathbf{B_{xz}} = \mathbf{B_{xy}}\mathbf{B_{yz}}$, where $[\mathbf{B_{xz}}]_{k,i}$ represents the overall influence of SNP $k$ on phenotype $i$ mediated by gene network $\mathbf{\Lambda_y}$ (Fig 1B, gray and tan dashed arrows). Such SNP effects on phenotypes are not directly modeled in the PerturbNet model but can be inferred by deriving the marginal distribution $p(\mathbf{z}|\mathbf{x})$ as follows:

$$p(\mathbf{z}|\mathbf{x}) = N(\mathbf{B_{xz}}^T\mathbf{x}, \ \mathbf{\Lambda_z}^{-1} + \mathbf{\Lambda_z}^{-1}\mathbf{\Theta_{yz}}^T\mathbf{\Lambda_y}^{-1}\mathbf{\Theta_{yz}}\mathbf{\Lambda_z}^{-1}).\tag{5}$$

From this, the marginal distribution of $[\mathbf{z}]_i$ for phenotype $i$ given $\mathbf{x}$ can be obtained as $p([\mathbf{z}]_i|\mathbf{x}) = N([[\mathbf{B_{xz}}]_{:,i}]^T\mathbf{x}, [\mathbf{\Lambda_z}^{-1} + \mathbf{\Lambda_z}^{-1}\mathbf{\Theta_{yz}}^T\mathbf{\Lambda_y}^{-1}\mathbf{\Theta_{yz}}\mathbf{\Lambda_z}^{-1}]_{i,i})$. Eq (5) can also be obtained from the marginalization of $\mathbf{y}$ in the joint distribution $p(\mathbf{z}, \mathbf{y}|\mathbf{x})$ derived from Eq (1):

$$p(\mathbf{z}, \mathbf{y}|\mathbf{x}) = N(-\mathbf{\Lambda_{zy}}^{-1}\mathbf{\Theta_{zy,x}}^T\mathbf{x}, \mathbf{\Lambda_{zy}}^{-1}),\tag{6}$$

where $\mathbf{\Theta_{zy,x}} = (\mathbf{0}_{p \times r}, \mathbf{\Theta_{xy}})$ with $\mathbf{0}_{p \times r}$ for $p \times r$ matrix of 0's and $\mathbf{\Lambda_{zy}} = \begin{pmatrix} \mathbf{\Lambda_z} & \mathbf{\Theta_{yz}^T} \\ \mathbf{\Theta_{yz}} & \mathbf{\Lambda_y} + \mathbf{\Theta_{yz}}\mathbf{\Lambda_z}^{-1}\mathbf{\Theta_{yz}^T} \end{pmatrix}$

This joint distribution is an alternative representation of the PerturbNet model in Eq (1) and corresponds to another sCGGM over **y** and **z** conditional on **x** with network $\mathbf{\Lambda_{zy}}$ and perturbation $\mathbf{\Theta_{zy,x}}$.

- **INF-II for inferring SNP effects on clinical phenotypes mediated by a gene module:** Let $M_1, \ldots, M_s$ be disjoint subsets of $q$ genes, where $\cup_{m=1,\ldots,s} M_m$ is the full set of $q$ genes (Fig 1B, gray and tan gene modules). Then, the overall SNP effects on phenotypes in $\mathbf{B_{xz}}$ can be decomposed into SNP effects on phenotypes mediated by each gene module $\mathbf{B_{xz}^{M_m}}$ as follows:

$$\mathbf{B_{xz}} = \sum_{m=1}^{s} \mathbf{B_{xz}^{M_m}},$$

where

$$\mathbf{B_{xz}^{M_m}} = \sum_{j \in M_m} [\mathbf{B_{xy}}]_{:,j}[\mathbf{B_{yz}}]_{j,:}.$$

In this decomposition, $[\mathbf{B_{xz}^{M_m}}]_{k,i}$ quantifies the effect of SNP $k$ on phenotype $i$ through the expression levels of genes in module $M_m$. If the module size is 1, $[\mathbf{B_{xz}^{M_m}}]_{k,i}$ corresponds to the SNP effect on the phenotype mediated by the single gene in the module.

- **INF-III for inferring posterior gene network after seeing phenotype data:** INF-III computes $\mathbf{\Lambda_{y|x,z}} = \mathbf{\Lambda_y} + \mathbf{\Theta_{yz}}\mathbf{\Lambda_z}^{-1}\mathbf{\Theta_{yz}^T}$ for gene network $\mathbf{\Lambda_y}$ augmented with the component $\mathbf{\Theta_{yz}}\mathbf{\Lambda_z}^{-1}\mathbf{\Theta_{yz}^T}$ (Fig 1B, blue dashed edge). In this augmented network, additional edges are introduced between two genes if their expression levels influence the same trait or if they both affect traits that are connected in the phenotype network $\mathbf{\Lambda_z}$. The posterior gene network $\mathbf{\Lambda_{y|x, z}}$ can be obtained by inferring the posterior distribution of expression levels given phenotypes from the estimated PerturbNet model:

$$p(\mathbf{y}|\mathbf{x}, \mathbf{z}) = N\left(-(\mathbf{z}^T\mathbf{\Theta_{yz}^T} + \mathbf{x}^T\mathbf{\Theta_{xy}})\mathbf{\Lambda_{y|x,z}}^{-1}, \; \mathbf{\Lambda_{y|x,z}}^{-1}\right).$$

This can also be seen from the alternative representation of the estimated model given as the joint distribution in Eq (6), in which $\mathbf{\Lambda_{y|x,z}}$ defines the network over **y** in the full network $\mathbf{\Lambda_{zy}}$. This process of introducing additional dependencies via $\mathbf{\Theta_{yz}}\mathbf{\Lambda_z}^{-1}\mathbf{\Theta_{yz}^T}$ in this sCGGM for joint distribution is known as moralization in the probabilistic graphical model literature [26].

## SNP perturbation effects on gene modules and trait groups

We summarize the estimated and inferred SNP perturbation effects at the module level and at the trait-group level, using a score function $S([\mathbf{A}]_{I,J}) = \sum_{i \in I, j \in J}|[\mathbf{A}]_{i,j}|$ for perturbation effects $\mathbf{A}$ of $R \times C$ matrix and for subsets of rows $I \subset \{1, \ldots, R\}$ and columns $J \subset \{1, \ldots, C\}$, as follows:

- **The effects of SNP $i$ on gene module $M$:** $S([\mathbf{\Theta_{xy}}]_{i,M})$ for direct effects and $S([\mathbf{B_{xy}}]_{i,M})$ for indirect effects. Similarly, we summarize the overall SNP effects on gene module $M$ as $S([\mathbf{\Theta_{xy}}]_{:,M})$ for direct effects and $S([\mathbf{B_{xy}}]_{:,M})$ for indirect effects.

- **The effects of gene module *M* on trait group *T*:** $S([\mathbf{\Theta_{yz}}]_{M,T})$ for direct effects and $S([\mathbf{B_{yz}}]_{M,T})$ for indirect effects.

- **The effects of SNP *i* on trait group *T*:** $S([\mathbf{B_{xz}}]_{i,T})$. Similarly, we use $S([\mathbf{B_{xz}}]_{:,T})$ to represent overall effects of SNPs on trait group *T*.

- **The effects of SNP *i* on trait group *T* mediated by module *M*:** $S([\mathbf{B_{xz}^{M}}]_{i,T})$. Overall SNP effects on trait group *T* mediated by module *M* is obtained as $S([\mathbf{B_{xz}^{M}}]_{:,T})$.

## PerturbNet learning algorithms

We introduce an efficient algorithm for obtaining a sparse estimate of the PerturbNet model parameters with few edges in the graph. Given genotype data $\mathbf{X} \in \mathbb{R}^{n \times p}$ for $n$ samples and $p$ SNPs, expression data $\mathbf{Y} \in \mathbb{R}^{n \times q}$ for $q$ genes, and phenotype data $\mathbf{Z} \in \mathbb{R}^{n \times r}$ for $r$ phenotypes, we estimate a sparse Gaussian chain graph model in Eq (1) by minimizing the negative log-likelihood of data along with sparsity-inducing $L_1$ penalty:

$$\min_{\mathbf{\Lambda_y} \succ 0, \mathbf{\Theta_{xy}}, \mathbf{\Lambda_z} \succ 0, \mathbf{\Theta_{yz}}} f(\mathbf{\Lambda_y}, \mathbf{\Theta_{xy}}) + f(\mathbf{\Lambda_z}, \mathbf{\Theta_{yz}}), \tag{7}$$

where

$$
\begin{aligned}
f(\mathbf{\Lambda_y}, \mathbf{\Theta_{xy}}) \quad &= -\log|\mathbf{\Lambda_y}| + \mathrm{tr}\,(\mathbf{S_{yy}}\mathbf{\Lambda_y} + 2\mathbf{S_{xy}}^T\mathbf{\Theta_{xy}} + \mathbf{\Lambda_y^{-1}}\mathbf{\Theta_{xy}^T}\mathbf{S_{xx}}\mathbf{\Theta_{xy}}) \\
&\quad + \lambda_{\mathbf{\Lambda_y}} \parallel \mathbf{\Lambda_y} \parallel_1 + \lambda_{\mathbf{\Theta_{xy}}} \parallel \mathbf{\Theta_{xy}} \parallel_1
\end{aligned}
\tag{8a}
$$

$$
\begin{aligned}
f(\mathbf{\Lambda_z}, \mathbf{\Theta_{yz}}) \quad &= -\log|\mathbf{\Lambda_z}| + \mathrm{tr}\,(\mathbf{S_{zz}}\mathbf{\Lambda_z} + 2\mathbf{S_{yz}}^T\mathbf{\Theta_{yz}} + \mathbf{\Lambda_z^{-1}}\mathbf{\Theta_{yz}^T}\mathbf{S_{yy}}\mathbf{\Theta_{yz}}) \\
&\quad + \lambda_{\mathbf{\Lambda_z}} \parallel \mathbf{\Lambda_z} \parallel_1 + \lambda_{\mathbf{\Theta_{yz}}} \parallel \mathbf{\Theta_{yz}} \parallel_1,
\end{aligned}
\tag{8b}
$$

given data covariance matrices $\mathbf{S_{xx}} = \frac{1}{n}\tilde{\mathbf{X}}^T\tilde{\mathbf{X}}$, $\mathbf{S_{xy}} = \frac{1}{n}\tilde{\mathbf{X}}^T\tilde{\mathbf{Y}}$, $\mathbf{S_{yy}} = \frac{1}{n}\tilde{\mathbf{Y}}^T\tilde{\mathbf{Y}}$, $\mathbf{S_{yz}} = \frac{1}{n}\tilde{\mathbf{Y}}^T\tilde{\mathbf{Z}}$, and $\mathbf{S_{zz}} = \frac{1}{n}\tilde{\mathbf{Z}}^T\tilde{\mathbf{Z}}$ for mean-centered data matrices $\tilde{\mathbf{X}}$, $\tilde{\mathbf{Y}}$, and $\tilde{\mathbf{Z}}$, and $\parallel \cdot \parallel_1$ for non-smooth element-wise $L_1$ penalty. The regularization parameters $\lambda_{\mathbf{\Lambda_y}}, \lambda_{\mathbf{\Theta_{xy}}}, \lambda_{\mathbf{\Lambda_z}}, \lambda_{\mathbf{\Theta_{yz}}} > 0$ are chosen to maximize the Bayesian information criterion (BIC). We do not penalize the diagonal entries of $\mathbf{\Lambda_y}$ and $\mathbf{\Lambda_z}$, following the common practice for sparse inverse covariance estimation. Solving Eq (7) is a convex optimization problem with a guarantee in finding the optimal solution.

To solve the PerturbNet estimation problem in Eq (7), we introduce Fast-sCGGM and Mega-sCGGM for an efficient sCGGM estimation to be used as a key subroutine. The problem in Eq (7) decouples into two subproblems, each containing one of two disjoint sets of parameters $\{\mathbf{\Lambda_y}, \mathbf{\Theta_{xy}}\}$ and $\{\mathbf{\Lambda_z}, \mathbf{\Theta_{yz}}\}$, each of which can be solved with an sCGGM optimization algorithm. Below, we introduce Fast-sCGGM for learning an sCGGM that substantially reduces computation time, compared to the previous state-of-the-art method NCD [25]. Then, we describe Mega-sCGGM, a modification of Fast-sCGGM, that performs block-wise computation to learn from a large dataset on a machine with limited memory. Empirically Fast-sCGGM and Meta-sCGGM led to orders-of-magnitude speedup compared to the existing state-of-the-art method.

**Fast-sCGGM for improving computation time.**   We discuss Fast-sCGGM for solving the sCGGM optimization problem in Eq (8a). The same approach can be used to solve Eq (8b). We drop the subscripts and use $\mathbf{\Theta}$ for $\mathbf{\Theta_{xy}}$ and $\mathbf{\Lambda}$ for $\mathbf{\Lambda_y}$ to simplify the notation. We re-write

the problem in Eq (8a) as

$$\min_{\mathbf{\Lambda} \succ 0, \mathbf{\Theta}} \quad f(\mathbf{\Lambda}, \mathbf{\Theta}) = g(\mathbf{\Lambda}, \mathbf{\Theta}) + h(\mathbf{\Lambda}, \mathbf{\Theta}), \tag{9}$$

where $g(\mathbf{\Lambda}, \mathbf{\Theta}) = -\log|\mathbf{\Lambda}| + \text{tr}(\mathbf{S_{yy}}\mathbf{\Lambda} + 2\mathbf{S_{xy}}^T\mathbf{\Theta} + \mathbf{\Lambda}^{-1}\mathbf{\Theta}^T\mathbf{S_{xx}}\mathbf{\Theta})$ is the smooth negative log-likelihood and $h(\mathbf{\Lambda}, \mathbf{\Theta}) = \lambda_{\mathbf{\Lambda}}\|\mathbf{\Lambda}\|_1 + \lambda_{\mathbf{\Theta}}\|\mathbf{\Theta}\|_1$ is the non-smooth elementwise $L_1$ penalty with regularization parameters $\lambda_{\mathbf{\Lambda}}$ and $\lambda_{\mathbf{\Theta}}$.

Fast-sCGGM uses an alternate Newton coordinate descent method and alternately updates $\mathbf{\Lambda}$ and $\mathbf{\Theta}$, optimizing Eq (9) over $\mathbf{\Lambda}$ given $\mathbf{\Theta}$ and vice versa until convergence. Our approach is based on the key observation that with $\mathbf{\Lambda}$ fixed, the problem of solving Eq (9) over $\mathbf{\Theta}$ becomes simply the well-known Lasso optimization, which can be solved efficiently using a coordinate descent method [32]. On the other hand, optimizing Eq (9) for $\mathbf{\Lambda}$ given $\mathbf{\Theta}$ requires forming a quadratic approximation to find a generalized Newton direction and performing line search to find a step size. However, this computation is significantly simpler than performing the same type of computation on both $\mathbf{\Lambda}$ and $\mathbf{\Theta}$ jointly as in the previous approach NCD [25]. Our algorithm iterates between the following two steps until convergence:

- **Coordinate descent optimization for $\mathbf{\Theta}$ given $\mathbf{\Lambda}$:** With $\mathbf{\Lambda}$ fixed, the optimization problem in Eq (9) becomes

$$\underset{\mathbf{\Theta}}{\arg\min} \ g_{\mathbf{\Lambda}}(\mathbf{\Theta}) + \lambda_{\mathbf{\Theta}}\| \mathbf{\Theta} \|_1, \tag{10}$$

where $g_{\mathbf{\Lambda}}(\mathbf{\Theta}) = \text{tr}(2\mathbf{S_{xy}}^T\mathbf{\Theta} + \mathbf{\Lambda}^{-1}\mathbf{\Theta}^T\mathbf{S_{xx}}\mathbf{\Theta})$. Since $g_{\mathbf{\Lambda}}(\mathbf{\Theta})$ is a quadratic function, Eq (10) corresponds to the Lasso problem and the coordinate descent method can be used to solve this efficiently.

- **Coordinate descent optimization for $\mathbf{\Lambda}$ given $\mathbf{\Theta}$:** With $\mathbf{\Theta}$ fixed, the problem in Eq (9) becomes

$$\underset{\mathbf{\Lambda} \succ 0}{\arg\min} \ g_{\mathbf{\Theta}}(\mathbf{\Lambda}) + \lambda_{\mathbf{\Lambda}}\| \mathbf{\Lambda} \|_1, \tag{11}$$

where $g_{\mathbf{\Theta}}(\mathbf{\Lambda}) = -\log|\mathbf{\Lambda}| + \text{tr}(\mathbf{S_{yy}}\mathbf{\Lambda} + \mathbf{\Lambda}^{-1}\mathbf{\Theta}^T\mathbf{S_{xx}}\mathbf{\Theta})$. To solve this, we first find a generalized Newton direction that minimizes the $L_1$-regularized quadratic approximation $\bar{g}_{\mathbf{\Lambda}, \mathbf{\Theta}}(\Delta_{\mathbf{\Lambda}})$ of $g_{\mathbf{\Theta}}(\mathbf{\Lambda})$:

$$\mathbf{D_{\Lambda}} = \underset{\Delta_{\mathbf{\Lambda}}}{\arg\min} \ \bar{g}_{\mathbf{\Lambda}, \mathbf{\Theta}}(\Delta_{\mathbf{\Lambda}}) + \lambda_{\mathbf{\Lambda}}\| \mathbf{\Lambda} + \Delta_{\mathbf{\Lambda}} \|_1, \tag{12}$$

where $\bar{g}_{\mathbf{\Lambda}, \mathbf{\Theta}}(\Delta_{\mathbf{\Lambda}})$ is obtained from a second-order Taylor expansion and is given as

$$\bar{g}_{\mathbf{\Lambda}, \mathbf{\Theta}}(\Delta_{\mathbf{\Lambda}}) = \ \text{vec}\,(\nabla_{\mathbf{\Lambda}} g(\mathbf{\Lambda}, \mathbf{\Theta}))^T \text{vec}\,(\Delta_{\mathbf{\Lambda}}) + \frac{1}{2}\,\text{vec}\,(\Delta_{\mathbf{\Lambda}})^T \nabla_{\mathbf{\Lambda}}^2 g(\mathbf{\Lambda}, \mathbf{\Theta})\,\text{vec}\,(\Delta_{\mathbf{\Lambda}}).$$

In the above equation, $\nabla_{\mathbf{\Lambda}}\, g(\mathbf{\Lambda}, \mathbf{\Theta}) = \mathbf{S_{yy}} - \mathbf{\Sigma} - \mathbf{\Psi}$ and $\nabla_{\mathbf{\Lambda}}^2 g(\mathbf{\Lambda}, \mathbf{\Theta}) = \mathbf{\Sigma} \otimes (\mathbf{\Sigma} + 2\mathbf{\Psi})$, where $\mathbf{\Sigma} = \mathbf{\Lambda}^{-1}$ and $\mathbf{\Psi} = \mathbf{\Sigma}\mathbf{\Theta}^T\mathbf{S_{xx}}\mathbf{\Theta}\mathbf{\Sigma}$, are the components of the gradient and Hessian matrices corresponding to $\mathbf{\Lambda}$. The problem in Eq (12) is again equivalent to the Lasso problem, which can be solved efficiently via coordinate descent. Given the Newton direction for $\mathbf{\Lambda}$, we update $\mathbf{\Lambda} \leftarrow \mathbf{\Lambda} + \alpha\, \mathbf{D_{\Lambda}}$, where step size $0 < \alpha \leq 1$ is set by line search on the objective in Eq (11) to ensure sufficient decrease in Eq (9) and positive definiteness of $\mathbf{\Lambda}$.

In order to further reduce computation time, we adopt the following strategies that have been previously used for sparse GGM and sCGGM optimizations [25, 54]. First, to improve

the efficiency of coordinate descent for the Lasso problem in Eqs (10) and (12), we restrict the updates to an active set of variables given as

$$\mathcal{S}_{\boldsymbol{\Lambda}} = \{[\Delta_{\boldsymbol{\Lambda}}]_{i,j} : |[\nabla_{\boldsymbol{\Lambda}} g(\boldsymbol{\Lambda}, \boldsymbol{\Theta})]_{i,j}| > \lambda_{\boldsymbol{\Lambda}} \vee [\boldsymbol{\Lambda}]_{i,j} \neq 0\}$$
$$\mathcal{S}_{\boldsymbol{\Theta}} = \{[\Delta_{\boldsymbol{\Theta}}]_{i,j} : |[\nabla_{\boldsymbol{\Theta}} g(\boldsymbol{\Lambda}, \boldsymbol{\Theta})]_{i,j}| > \lambda_{\boldsymbol{\Theta}} \vee [\boldsymbol{\Theta}]_{i,j} \neq 0\}.$$

Because the active set sizes $m_{\boldsymbol{\Lambda}} = |\mathcal{S}_{\boldsymbol{\Lambda}}|$, $m_{\boldsymbol{\Theta}} = |\mathcal{S}_{\boldsymbol{\Theta}}|$ approach the number of non-zero entries in the sparse solutions for $\boldsymbol{\Lambda}$ and $\boldsymbol{\Theta}$ over iterations, this strategy yields a substantial speedup. Second, to further improve the efficiency of coordinate descent, we store intermediate results for the large matrix products that need to be computed repeatedly. We compute and store $\mathbf{U} := \Delta_{\boldsymbol{\Lambda}}\Sigma$ and $\mathbf{V} := \Delta_{\boldsymbol{\Theta}}\Sigma$ at the beginning of the optimization. Then, after a coordinate descent update to $[\Delta_{\boldsymbol{\Lambda}}]_{i,j}$, rows $i$ and $j$ of $\mathbf{U}$ are updated. Similarly, after an update to $[\Delta_{\boldsymbol{\Theta}}]_{i,j}$, row $i$ of $\mathbf{V}$ is updated. Finally, in each iteration, we warm-start $\boldsymbol{\Lambda}$ and $\boldsymbol{\Theta}$ from the results of the previous iteration and make a single pass over the active set. This ensures decrease in the objective in Eq (9), while reducing the overall computation time in practice. The pseudocode for Fast-sCGGM is provided in S2 Text.

**Mega-sCGGM for removing memory constraint.** Fast-sCGGM as described above is still limited by the space required to store large matrices during coordinate descent computation. Solving Eq (12) for updating $\boldsymbol{\Lambda}$ requires precomputing and storing $q \times q$ matrices, $\Sigma$ and $\Psi = \Sigma\boldsymbol{\Theta}^T\mathbf{S}_{\mathbf{xx}}\boldsymbol{\Theta}\Sigma$, and solving Eq (10) for updating $\boldsymbol{\Theta}$ requires $\Sigma$ and a $p \times p$ matrix $\mathbf{S}_{\mathbf{xx}}$. A naive approach to reduce the memory footprint would be to recompute portions of these matrices on demand for each coordinate update, which would be very expensive.

Here, we describe Mega-sCGGM that combines the alternating Newton coordinate descent algorithm in Fast-sCGGM with block coordinate descent to scale up the optimization to very large problems on a machine with limited memory. During coordinate descent optimization, Mega-sCGGM updates blocks of $\boldsymbol{\Lambda}$ and $\boldsymbol{\Theta}$ so that within each block, the computation of the large matrices can be cached and re-used. These blocks are determined automatically by exploiting the sparse stucture. For $\boldsymbol{\Lambda}$, we extend the block coordinate descent approach in BIG&QUIC [31] developed for sparse GGMs to take into account the conditioning variables in sCGGMs. For $\boldsymbol{\Theta}$, we describe a new approach for block coordinate descent update. The block-wise update in Mega-sCGGM is described in detail in S2 Text and illustrated in S8 Fig. Our algorithm can, in principle, be applied to problems of any size on a machine with limited memory.

The previous method NCD, Fast-sCGGM, and Mega-sCGGM solve the same optimization problem and produce nearly identical estimate. However, Fast-sCGGM reaches this estimate more efficiently than the NCD and Mega-sCGGM finds the estimate without running out of memory.

**Parallelization in Fast-sCGGM and Mega-sCGGM.** We parallelize some of the expensive computations in Fast-sCGGM and Mega-sCGGM on multi-core machines. For both methods, we parallelize all matrix-matrix and matrix-vector multiplications. In addition, we parallelize the computation of columns of $\Sigma$ and $\Psi$ in Fast-sCGGM and the same computation within each block in Mega-sCGGM. In Mega-sCGGM, we parallelize the computation of each row of $\mathbf{S}_{\mathbf{xx}}$ whenever it is recomputed.

**Semi-supervised learning.** We introduce a modification of our learning algorithm for semi-supervised learning, to handle the situation where expression data are available only for a subset of individuals because of the difficulty of obtaining tissue samples. Our strategy is to use an EM algorithm [55] that imputes the missing expression levels in the E-step and performs our Fast-sCGGM or Mega-sCGGM optimization in the M-step. Given a dataset $\mathcal{D} = \{\mathcal{D}_o, \mathcal{D}_h\}$, where $\mathcal{D}_o = \{\mathbf{X}_o, \mathbf{Y}_o, \mathbf{Z}_o\}$ for the fully-observed data and $\mathcal{D}_h = \{\mathbf{X}_h, \mathbf{Z}_h\}$ for the

samples with missing gene-expression levels, our EM algorithm [55] maximizes the expected log-likelihood of data:

$$\mathcal{L}(\mathcal{D}_o; \boldsymbol{\Theta}) + \mathrm{E}[\mathcal{L}(\mathcal{D}_h, \mathbf{Y}_h; \boldsymbol{\Theta})],$$

combined with $L_1$-regularization, where $\mathcal{L}(\mathcal{D}_o; \boldsymbol{\Theta})$ and $\mathcal{L}(\mathcal{D}_h; \boldsymbol{\Theta})$ are the log-likelihood of data $\mathcal{D}_o$ and $\mathcal{D}_h$ with respect to the model in Eq (1) and the expectation is taken with respect to:

$$p(\mathbf{y}|\mathbf{z}, \mathbf{x}) = N(\mu_{\mathbf{y}|\mathbf{x},\mathbf{z}}, \Sigma_{\mathbf{y}|\mathbf{x},\mathbf{z}}),$$

$$\mu_{\mathbf{y}|\mathbf{x},\mathbf{z}} = -\Sigma_{\mathbf{y}|\mathbf{x},\mathbf{z}}(\boldsymbol{\Theta}_{\mathbf{yz}}\mathbf{z} + \boldsymbol{\Theta}_{\mathbf{xy}}^T\mathbf{x}) \quad \text{and} \quad \Sigma_{\mathbf{y}|\mathbf{x},\mathbf{z}} = (\boldsymbol{\Lambda}_{\mathbf{y}} + \boldsymbol{\Theta}_{\mathbf{yz}}\boldsymbol{\Lambda}_{\mathbf{z}}\boldsymbol{\Theta}_{\mathbf{yz}}^T)^{-1}. \tag{13}$$

In E-step, a naive inversion of $\boldsymbol{\Lambda}_{\mathbf{y}} + \boldsymbol{\Theta}_{\mathbf{yz}}\boldsymbol{\Lambda}_{\mathbf{z}}^{-1}\boldsymbol{\Theta}_{\mathbf{yz}}^T$ to obtain $\Sigma_{\mathbf{y}|\mathbf{x},\mathbf{z}}$ is expensive and requires a storage for this dense matrix that may exceed computer memory for large gene expression datasets. To make the EM algorithm efficient in terms of both time and memory, we assume that the number of phenotypes $r$ is relatively small compared to the number of genes (i.e., $r \ll q$), which is typical for most studies. Then, instead of explicitly performing the E-step, we embed the E-step within the M-step by representing the E-step results implicitly to fit in memory and computing them explicitly on-demand as needed in the M-step. Specifically, we implicitly represent $\boldsymbol{\Lambda}_{\mathbf{y}} + \boldsymbol{\Theta}_{\mathbf{yz}}\boldsymbol{\Lambda}_{\mathbf{z}}^{-1}\boldsymbol{\Theta}_{\mathbf{yz}}^T$ as $\boldsymbol{\Lambda}_{\mathbf{y}} + \mathbf{KK}^T$, using low-rank component $\mathbf{K} = \boldsymbol{\Theta}_{\mathbf{yz}}\mathbf{L}_{\mathbf{z}}^T$ and the sparse Cholesky factorization of trait network $\mathbf{L}_{\mathbf{z}}\mathbf{L}_{\mathbf{z}}^T = \boldsymbol{\Lambda}_{\mathbf{zz}}$. Then, during M-step, we invert $\boldsymbol{\Lambda}_{\mathbf{y}} + \mathbf{KK}^T$, one column at a time as needed, using the conjugate gradient method. This modified EM algorithm produces the same estimate as the original EM algorithm that iterates between an M-step and an E-step.

## Simulation experiments

**Simulated data.**    To assess the accuracy of the computational methods, we generated simulated data from the known ground-truth models and SNPs in the CAMP data. For each type of models, we obtained sensitivities at different FDRs averaged over 10 simulated datasets. To assess accuracy on the recovery of trait-perturbing SNPs and mediator genes, we simulated data from two types of ground-truth models: a two-layer sCGGM as in PerturbNet to mimic the SNP perturbations of clinical traits modulated by gene networks and a two-layer linear regression model to mirror the assumptions of mediator genes acting independently of other genes as in the existing methods. To assess the accuracy of gene network learning, we again simulated data from two types of ground-truth models: a two-layer sCGGM with a gene network perturbed by SNPs and a two-layer GGM [56] with no perturbation of networks.

We set the ground-truth two-layer sCGGM as follows. We assumed gene network $\boldsymbol{\Lambda}_{\mathbf{y}}$ of size $q = 5,000$ and trait network $\boldsymbol{\Lambda}_{\mathbf{z}}$ of size $r = 100$. In both networks, we assumed 50 nodes in each module, leading to 100 modules in $\boldsymbol{\Lambda}_{\mathbf{y}}$ and two modules in $\boldsymbol{\Lambda}_{\mathbf{z}}$. Then, the algorithm described in [57] was used to generate scale-free networks with average degree 3 and with 90% of the edges within modules and 10% of the edges across modules. Edge weights were randomly generated from normal distribution $\mathcal{N}(0.5, 0.1^2)$. The diagonal elements of $\boldsymbol{\Lambda}_{\mathbf{z}}$ and $\boldsymbol{\Lambda}_{\mathbf{y}}$ were set to ensure the minimum eigenvalue of the matrix is 0.3. The perturbation parameters $\boldsymbol{\Theta}_{\mathbf{yz}}$ and $\boldsymbol{\Theta}_{\mathbf{xy}}$ were set to represent perturbation of modules. For $\boldsymbol{\Theta}_{\mathbf{yz}}$, 10 out of 50 gene modules were set to have no influence on the traits. Each half of the other 40 modules were set to influence primarily each of the two trait modules, 90% of the edges connecting to one trait module and 10% to the other trait module. For $\boldsymbol{\Theta}_{\mathbf{xy}}$, 25 gene modules were perturbed by 1,000 randomly-selected SNPs and the other 25 modules were not perturbed by any SNPs. In both perturbation parameters, each expression and clinical trait was perturbed by three perturbants.

The magnitudes of perturbations were sampled from normal distribution $\mathcal{N}(0.5, 0.1^2)$, with edge signs randomly assigned with probability 0.5 to positive or negative.

For simulation from a two-layer linear regression model, the regression parameters were set using the same approach as $\mathbf{\Theta_{xy}}$ and $\mathbf{\Theta_{yz}}$ above and the noise variances were set to 0.5. For comparison on the task of gene network recovery, we generated data from a model assuming no SNP perturbations, using a two-layer GGM [30] with the same parameters as in the sCGGM network parameters.

**Comparison of methods on simulated data.** Using simulated data, we compared the accuracy of PerturbNet, PrediXcan [12], eCAVIAR [13], and two-layer Lasso (S1 Text) on the tasks of identifying SNPs perturbing clinical traits and identifying genes mediating these perturbations. PrediXcan tests for an association between a trait and an imputed gene expression value obtained as an elastic-net prediction given SNPs, providing a list of genes mediating overall genetic effects on each trait ($p$-value $< 0.01$). From these mediator genes for overall SNP effects, we obtained the mediator genes for an individual SNP by further weighting $-\log$ ($p$- value) for the given gene/trait pair with the elastic-net regression coefficient for each SNP/gene pair. To obtain SNPs with the strongest effects on each trait, we further aggregated these scores by summing over all genes for each SNP/trait pair. eCAVIAR assigns a colocalization posterior probability (CLPP) score, the posterior probability of a GWAS SNP and an eQTL in the same genomic region of $M$ SNPs co-localizing, to each triplet of SNP, gene, and trait, given statistically significant GWAS SNPs and eQTLs (FDR $q < 10^{-3}$). To find SNPs with the strongest effects on traits, we collapsed the CLPP scores for triplets into scores for SNP/trait pairs by taking the maximum CLPP score across all genes for each SNP/trait pair. Then, we identified the corresponding gene as the strongest mediating gene for each SNP/trait pair. To identify genes that are the strongest mediators of overall SNP effects on a trait, we collapsed the CLPP scores for triplets into scores for gene/trait pairs by summing CLPP scores over all SNPs for the given/trait pair and selected the highest-scoring gene. For PerturbNet and two-layer Lasso, we selected the regularization parameters that maximize the BIC.

### Analysis of CAMP asthma data

**Asthma dataset.** We preprocessed the genotype, gene expression, and clinical phenotype data, collected from asthma patients participating in the CAMP study [27–29] for our analysis. We used 174 non-Hispanic Caucasian subjects for whom both genotype and clinical phenotype data were available. For a subset of 140 individuals, gene expression data from primary peripheral blood CD4+ lymphocytes were also available. After removing SNPs with minor allele frequency less than 0.1 and those with missing reference SNP ids, we obtained 495,597 SNPs for autosomal chromosomes. Given expression levels for 22,184 mRNA transcripts profiled with Illumina HumanRef8 v2 BeadChip arrays [29], we removed transcript levels with expression variance less than 0.01, which resulted in the 11,598 transcript levels to be used in our analysis. Then, we converted the expression values to their $z$-scores. The clinical phenotype data comprised 35 phenotypes (S3 Table), including 25 features related to lung function and 10 features collected via blood testing. The phenotypes were verified to be well-approximated by normal distributions: with the exception of rescue_bd_7day and prevent_bd_7day, which are counts and skewed right, all the other variables represented continuous measurements distributed roughly symmetrically in quantile-quantile (QQ) plots (S7 Fig). The clinical phenotypes were converted to their $z$-scores within each phenotype so that all phenotypes have equal variance. We then imputed missing values using low-rank matrix completion [58].

**Comparison of methods on asthma data.** We compared PerturbNet with PrediXcan, eCAVIAR, and two-layer Lasso on the asthma data. We applied PerturbNet with semi-

supervised learning to all samples, including the samples with only genotype and clinical trait data. In PerturbNet and two-layer Lasso, we selected the regularization parameters using the BIC. With PrediXcan, we mimiced semi-supervised learning by fitting elastic net to the samples with both expression and genotype data, imputing the missing expression data, and using all samples to perform association tests between the imputed gene expression values and traits. To obtain top trait-associated SNPs, we scored each SNP by first scoring SNP-trait pairs as in the simulation study above and summing these scores across all traits. For eCAVIAR, GWAS SNPs were found using all samples, while eQTLs were found using only the samples with expression data. Then, top asthma-associated SNPs were found by summarizing CLPP scores of all triplets into scores of each SNP by taking maximum over all mediator genes and summing over all traits. We applied the two-layer Lasso only to the samples with all data and obtained top SNPs perturbing asthma traits using the same strategy as in PerturbNet.

**Comparison of computation time.** To assess the computation time of Mega-sCGGM, PerturbNet, and other existing methods, we used the same hardware setup with comparable software implementations. For Lasso, we used the implementation in GLMNET [59] with a backend written in Fortran. For NCD, we took the implementation written in C++ provided by the authors [25] and sped up this implementation with the Eigen matrix library [60], by employing low-rank matrix representations and using sparse matrix multiplications. For Mega-sCGGM and NCD, we used the same regularization parameters to ensure the resulting estimates are identical, and for Lasso, we chose the regularization parameters so that the $L_1$-norm of the regression coefficients roughly matched that of inferred indirect SNP effects in PerturbNet. We used the C++ implementation of eCAVIAR provided by the authors [13], and used PrediXcan [12] with elastic-net as implemented in GLMNET [59]. For all methods, the code was compiled and run with OpenMP multi-threading enabled on the same machines with 20Gb memory and 16 cores.

**Comparison of SNPs with external annotations.** For comparison with the DNase hypersensitivity sites, we used the UES tool [61] to identify the sites enriched in top asthma SNPs from each method and to compute the enrichment $p$-values for these overlaps with 100 Monte Carlo simulations. For comparison with annotations in RegulomeDB [36], we scored the top 500 asthma SNPs from each method into six categories: category 1 for overlaps with previously reported eQTLs with additional functional annotation on TF binding; categories 2-5 for overlaps with TF binding sites in ChIP-seq, DNase-seq, and motif hits, where lower scores indicate stronger evidence for being functional; and category 6 for little evidence of being functional. Finally, we compared the asthma SNPs found by each method with the previously reported asthma-associated SNPs in the NHGRI GWAS catalog [37]. Out of 557 SNPs in the NHGRI GWAS catalog for asthma, we examined how many of these SNPs are within 10, 20, 50, and 100kb of top $k$ SNPs from each method, $k$ ranging from 1 to 500. To assess the statistical significance of the overlap between two sets of SNPs, we performed a permutation test: for top $k$ SNPs from the given method, we generated 10,000 random sets of $k$ SNPs from the SNPs employed in the analysis to find the distribution of overlaps under the null hypothesis and reported the overlaps with $p$-value $< 0.05$.

## Supporting information

**S1 Text. Two-layer Lasso.**
(PDF)

**S2 Text. Fast-sCGGM and Mega-sCGGM for efficient sCGGM optimization.**
(PDF)

**S1 Fig. The PerturbNet model estimated from the CAMP data.**
(PDF)

**S2 Fig. The PerturbNet posterior gene network for asthma.**
(PDF)

**S3 Fig. SNP effects on gene modules and gene-module effects on phenotypes in asthma determined by PerturbNet.**
(PDF)

**S4 Fig. Manhattan plots for overall SNP effects on asthma phenotypes determined by PerturbNet.**
(PDF)

**S5 Fig. PerturbNet asthma SNPs as eQTLs.**
(PDF)

**S6 Fig. Overlap with asthma SNPs in NHGRI GWAS catalog.**
(PDF)

**S7 Fig. The distribution of clinical phenotypes in the CAMP data.**
(PDF)

**S8 Fig. Schematic of block coordinate descent in Mega-sCGGM.**
(PDF)

**S1 Table. GO categories enriched in gene modules in the PerturbNet asthma gene network.**
(PDF)

**S2 Table. GO categories enriched in genes in modules 16-20 connected to modules 13-15 in the PerturbNet asthma posterior gene network.**
(PDF)

**S3 Table. Description of asthma clinical phenotypes in CAMP data.**
(PDF)

## Acknowledgments

The authors thank Kathryn Roeder for helpful discussion.

## Author Contributions

**Conceptualization:** Calvin McCarter, Seyoung Kim.

**Data curation:** Calvin McCarter, Judie Howrylak.

**Formal analysis:** Calvin McCarter, Seyoung Kim.

**Funding acquisition:** Seyoung Kim.

**Investigation:** Calvin McCarter, Seyoung Kim.

**Methodology:** Calvin McCarter, Seyoung Kim.

**Project administration:** Calvin McCarter, Seyoung Kim.

**Resources:** Calvin McCarter.

**Software:** Calvin McCarter.

**Supervision:** Seyoung Kim.

**Visualization:** Calvin McCarter.

**Writing – original draft:** Calvin McCarter, Seyoung Kim.

**Writing – review & editing:** Calvin McCarter, Judie Howrylak, Seyoung Kim.

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
