## [Decision Letter · Decision Letter 0]

9 Jan 2020

Dear Dr Kim,

Thank you very much for submitting your manuscript 'Learning Gene Networks Underlying Clinical Phenotypes Using SNP Perturbation' for review by PLOS Computational Biology. We apologize for the rather lengthy initial review process, as finding reviewers with the required expertise took some time. 

Your manuscript has now been fully evaluated by the PLOS Computational Biology editorial team and by independent peer reviewers. The reviewers appreciated the attention to an important problem, but raised some substantial concerns about the manuscript as it currently stands. While your manuscript cannot be accepted in its present form, we are willing to consider a revised version in which the issues raised by the reviewers have been adequately addressed. 

If you have any concerns or questions, please do not hesitate to contact us.

Sincerely,

Christina S. Leslie

Associate Editor

PLOS Computational Biology

William Noble

Deputy Editor

PLOS Computational Biology

[LINK]

Reviewer's Responses to Questions

**Comments to the Authors:**

Reviewer #1: This paper introduces a method called PerturbNet, which can integrate SNP, gene expression and trait information in a single graphical model that can be used to infer the gene network mediated effect of SNPs on the traits. Furthermore, it can decompose the impact of different SNPs on different network components and also infer the underlying network. The model is based on a sparse Conditional Gaussian graphical model (sCGGM). The authors also introduce a new algorithm for learning the scGGM, which is significantly faster than the current implementations. They apply their approach to simulated data and also to data from the Childhood Asthma Management Program (CAMP) and also compare to other methods

such PrediXcan, eCAVIAR and a two-layered LASSO and show that their method is much better for identifying the relevant snps. Overall, this is an interesting piece of work and the PerturbNet method should be helpful to analyze datasets from consortia where all three types of measurements are being made. My only main comments to improve the work are (a) a better description of the method (b) comparisons to related work, (c) availability of PerturbNet. Details follow:

1. The PerturbNet approach is based on a sparse CGGM model, which aims to model p(y,z|x) as p(y|x)p(z|y) where x is the SNP information, y is the expre ssion and z is the phenotypic traits. However, fig 1 does not provide any description about the components of the sCGGM approach that is being used. A more annotated version of fig 1 which introduces the different components of the model would be very helpful to the reader. The manuscript text could also go into more details of the methods rather than pushing everything to methods.

2. The simulation makes use of a GGM and CGGM model and hence it is not surprising that PerturbNet works very well here. It might be good to discuss the simulation model in the content of the modeling assumptions of the different methods.

3. All the methods compared for linking SNPs to traits via networks are not network based. There have been other pieces of work that aim to identify subnetworks that are perturbed or explain the impact of the SNPs on phenotypes by a network. For example, the work from https://www.ncbi.nlm.nih.gov/pmc/articles/PMC4835676/, https://www.ncbi.nlm.nih.gov/pubmed/25781485, https://www.ncbi.nlm.nih.gov/pubmed/21390271. Comparison of PerturbNet to some network-based methods would be very beneficial.

4. The module identification step is not discussed in much detail. It is unclear how the number of modules is defined, why it is needed and if and how the module information is leveraged for network inference. It seems that it is used primarily as a post-processing step. It is also not clear how the modules are defined, is it on the network structure inferred or is it on the precision matrix or some other weighted graph.

5. The number of samples in the dataset seems low. It is unclear if the authors can assert any confidence on the inferred model. The FDR is mentioned but it is unclear how it is estimated.

6. The code for PerturbNet is not available. This is important to make available as soon as possible.

7. The Gaussian assumption has some nice properties which make the inference tractable. But it might be worth discussing if the variables, namely the traits are Gaussian.

8. The new Mega-sCGGM method seems a nice contribution of this work. Code for this should be made available. It would also be helpful to see if this more efficient version has the same or different performance as the original/exisiting implementations of CGMMs.

9. I might have missed this, but there must be hyperparamters to control for sparsity, but I did not see how this was set.

Minor:

1. Please define CLPP on page 12

2. The number of modules in the simulation is not clear. The sentence "..were assumed to have modules of size 50 each, leading to 100 and 2 442

modules in each network." needs revision (line 442).

Reviewer #2: McCarter et al. focus on an important research problem of identifying SNPs that influence phenotypes via molecular networks. While a number of computational methods exist for various GWAS and eQTL analysis, very few methods, if any, exist that connect genotype to phenotypes via transcriptional networks.

Authors propose a model that builds on probabilistic graphical models, in particular (multivariate) conditional Gaussian graphical model (or Markov random field). The model is constructed as a chain-structured graph in such a way that transcriptional level (or network) is conditioned with genotype and phenotype level is conditioned by the transcriptional networks. This construction allows considering perturbations in the genotype to affect phenotypes via the intermediate transcriptional networks. Authors propose a sparsity promoting optimisation method for the model including all layers using the standard L1 regularizers. Authors also propose three inference procedures to study the learned model to gain more detailed understanding of how genotype affects phenotypes via transcriptional networks. The model is coherently defined in the probabilistic graphical modeling framework. The model is well-motivated, although it is built on the assumption of all variables having a joint Gaussian distribution. Despite the joint normal assumption, the method can reveal reasonable results and probably serves as a good first approximation. The model and methods are clearly described and seems technically correct. Results demonstrate good performance relative to the state-of-the-art.

Major comments:

- page 9: The model space is extremely large while only limited amounts of data is typically available. Although the model learning relies on standard regularisation methods, I am worried that the model learning can be very sensitive. Authors should demonstrate the robustness (or sensitivity) or they model learning with an appropriate robustness analysis. E.g. how much does the learned model/network structure differ for small perturbations in the input data? How much do these changes affect the interpretation of the model (ref. the proposed 3 inference methods)? Are the results in Results section sensitive to small perturbations in the training data? I am OK with the method being marginally sensitive, as long as authors give a comprehensive evaluation of their methods, and give a fair discussion of potential limitations as well.

- page 12: For simulation studies, data is generated from Gaussian distributions. Can you carry out simulated experiments where data is generated from a model that includes nonlinearities?

- page 12: Authors should describe that how realistic the simulation set-up is? Or mention that data is simulated from a model that does not necessarily represent a realistic system.

Minor comments:

- page 7, row 282: A reader can easily misinterpret that x is a three-valued vector of length p, but text says "given as minor allele frequencies". Can you define the domain of x similarly as you do for y and z.

- page 9, row 331: I understood that the speed improvement (of several orders of magnitude) is based on *empirical* analysis. Please explicitly clarify that in the text too.

- pages 10-11: Could authors specify and write the joint Gaussian distribution for the (whole) model from which the marginals (e.g. Eq. On row 381-382) could be obtained using standard marginalisation techniques?

**Have all data underlying the figures and results presented in the manuscript been provided?**

Reviewer #1: Yes

Reviewer #2: Yes

PLOS authors have the option to publish the peer review history of their article (what does this mean?). If published, this will include your full peer review and any attached files.

Reviewer #1: No

Reviewer #2: No

---

## [Decision Letter · Decision Letter 1]

11 May 2020

Dear Dr. Kim,

We are pleased to inform you that your manuscript 'Learning Gene Networks Underlying Clinical Phenotypes Using SNP Perturbation' has been provisionally accepted for publication in PLOS Computational Biology.

Best regards,

Christina S. Leslie

Associate Editor

PLOS Computational Biology

William Noble

Deputy Editor

PLOS Computational Biology

Reviewer's Responses to Questions

**Comments to the Authors:**

Reviewer #1: The authors have addressed my comments.

Reviewer #2: The authors have addressed all my comments. (Upon rereading my previous comments I noticed a typo in one of my comments, but authors had appropriately addressed the comment anyways.)

**Have all data underlying the figures and results presented in the manuscript been provided?**

Reviewer #1: Yes

Reviewer #2: Yes

PLOS authors have the option to publish the peer review history of their article (what does this mean?). If published, this will include your full peer review and any attached files.

Reviewer #1: No

Reviewer #2: No

---

## [Editor Report · Acceptance letter]

6 Oct 2020

PCOMPBIOL-D-19-01547R1 

Learning Gene Networks Underlying Clinical Phenotypes Using SNP Perturbation

Dear Dr Kim,

I am pleased to inform you that your manuscript has been formally accepted for publication in PLOS Computational Biology. Your manuscript is now with our production department and you will be notified of the publication date in due course.

With kind regards,

Sarah Hammond
